analytical chemistry/biochemistry

optical force, vesicles, gold nanoparticle

**Author for correspondence:**
Takashi Kaneta
e-mail: kaneta@okayama-u.ac.jp

This article has been edited by the Royal Society of Chemistry, including the commissioning, peer review process and editorial aspects up to the point of acceptance.

# Enhancement of optical force acting on vesicles via the binding of gold nanoparticles

## Yumeki Tani and Takashi Kaneta

Department of Chemistry, Graduate School of Natural Science and Technology, Okayama University, Okayama 700-8530, Japan

YT, 0000-0002-6004-7973; TK, 0000-0001-9076-3906

Here we found that gold nanoparticles (AuNPs) enhance the optical force acting on vesicles prepared from phospholipids via hydrophobic and electrostatic interactions. A laser beam was introduced into a cuvette filled with a suspension of vesicles and it accelerated them in its propagation direction via a scattering force. The addition of the AuNPs exponentially increased the velocity of the vesicles as their concentration increased, but polystyrene particles had no significant impact on velocity in the presence of AuNPs. To elucidate the mechanism of the increased velocity, the surface charges in the vesicles and the AuNPs were controlled; the surface charges of the vesicles were varied via the use of anionic, cationic and neutral phospholipids, whereas AuNPs with positive and negative charges were synthesized by coating with citrate ion and 4-dimethylaminopyridine, respectively. All vesicles increased the velocity at different degrees depending on the surface charge. The vesicles were accelerated more efficiently when their charges were opposite those of the AuNPs. These results suggested that hydrophobic and electrostatic interactions between the vesicles and the AuNPs enhanced the optical force. By accounting for the binding constant between the vesicles and the AuNPs, we proposed a model for the relationship between the concentration of the AuNPs and the velocity of the vesicles. Consequently, the increased velocity of the vesicles was attributed to the light scattering that was enhanced when AuNPs were adsorbed onto the vesicles.

## 1. Introduction

In biological systems, vesicles play important roles that include the transportation of chemical substances into or out of cells, the storage of biosynthetic products and the digesting of nutrients [1–3]. The vesicles mainly consist of phospholipids that form lipid bilayers to isolate the cytoplasmic matrix from the surrounding medium. When vesicles fuse with cells, similar

constituents in the vesicle and cell membranes allow the transfer of content to cells. Conversely, the cells release the extracellular vesicles, which includes exosomes, microvesicles and apoptotic bodies. Extracellular vesicles have attracted much attention for uses in drug delivery, diagnosis and cell-to-cell communication [4–9].

Exosomes have diameters that range between 30 and 100 nm, and they facilitate communication between cells that effectively controls the immune response [10,11]. In addition, the exosomes are expected to be novel biomarkers for cancers because they contain information from the cells that release them [12–14]. The latest cancer research has revealed that extracellular vesicles are involved in all stages of cancer development [15]. Research has also shown that the exosomes released from the airways of epithelial cells are one of the factors in chronic obstructive pulmonary disease [16].

Obviously, the efficient isolation of exosomes is essential for clarifying their roles and functions. Ultracentrifugation is widely used for the isolation of exosomes [17–19], even though it is time-consuming and results in low recovery. Also, there is the necessity of an expensive reagent and a large sample amount. Microfluidic devices have achieved high rates of recovery, but these require sophisticated fabrication techniques and complex pumping systems [20–22]. Compared with these separation methods, manipulation using optical force, which is generally employed in techniques such as laser trapping or the use of optical tweezers [23], is useful for the collection and manipulation of small droplets and vesicles [24–27]. Optical force permits the trapping of objects such as particles or cells via the focus of a laser beam. Optical force has non-contact and non-destructive characteristics, which makes it suitable for applications in medical and biological sciences.

Recently, we reported that micro- and nanovesicles could be collected on a glass substrate using optical force generated by a laser beam [28]. Furthermore, the addition of gold nanoparticles (AuNPs) significantly improved the collection efficiency, and reduced collection time by a factor of 10. In that study, thermal convection due to light absorption of free AuNPs was observed in a solution. Consequently, the enhanced collection speed was attributed to thermal convection and possible enhancement of light scattering, but the mechanism remained unclear with respect to the interaction between the vesicles and the AuNPs that may have influenced the light scattering, that is, the optical force acting on the vesicle.

In the present study, we elucidated the mechanism by which the interaction between the vesicles and the AuNPs enhances the acceleration of the vesicles by optical force. Effects of the surface charges for the AuNPs and the vesicles were investigated by synthesizing two types of AuNPs and by preparing three types of vesicles. Citrate-AuNPs with negative charges and 4-dimethylaminopyridine (DMAP)-AuNPs with positive charges were synthesized, whereas the surface charges of the vesicles were controlled using cationic and anionic phospholipids that were mixed with neutral one. Various sizes of DMAP-AuNPs were also synthesized in order to investigate the effect that size exerted on the acceleration of the vesicle. We revealed that acceleration is enhanced when AuNPs bind to vesicles via electrostatic and hydrophobic interactions.

# 2. Experimental

## 2.1. Materials

All reagents used in this study were of analytical grade. Deionized water was prepared using an Elix water purification system (Millipore Co. Ltd, Molsheim, France). Dipalmitoylphosphatidylcholine (DPPC), 1,2-dipalmitoyl-3-trimethylammonium-propane chloride salt (TAP) and 1,2-dipalmitoyl-$sn$-glycero-3-phosphate sodium salt (PA) were obtained from Avanti Polar Lipids (Alabaster, AL, USA). Phosphate buffered saline (PBS) was purchased from Thermo Fisher Scientific (Yokohama, Japan). Acetone, chloroform, toluene, sodium sulphate anhydrous, sodium borohydride, rhodamine B, sodium tetrachloroaurate(III) dihydrate, 4-dimethylaminopyridine (DMAP) and tetraoctylammonium bromide were purchased from Wako Pure Chemical Industries (Osaka, Japan). Ethanol was purchased from Sigma-Aldrich (St Louis, MO, USA). Sulphuric acid, hydrochloric acid, aqueous ammonia and 30% hydrogen peroxide were obtained from Kanto Chemical (Tokyo, Japan). Polystyrene beads (diameter, 1.0 µm) were purchased from Funakoshi (Tokyo, Japan).

## 2.2 Preparation of vesicles

The vesicles with neutral charges were prepared with DPPC according to a method reported in previous research [28]. According to the protocol recommended by the supplier, PA was dissolved in a mixture of chloroform, methanol and aqueous ammonia at a ratio of 60.2 : 32.4 : 7.4. The vesicles with positive or

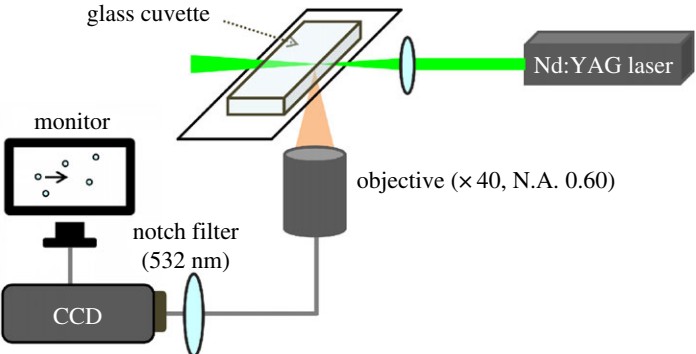

**Figure 1.** Schematic illustration of the optical system for measuring the velocity of the vesicle.

negative charges were prepared by mixing the DPPC solution with a TAP or PA solution at a ratio of 10 : 1 using the same protocol as that for neutral vesicles. A further increase in the content of the charged lipids prevented the formation of vesicles due to electrostatic repulsion. The sizes of the vesicles were adjusted to 1 μm via passage through an extruder.

## 2.3. Synthesis of citrate- and DMAP-AuNPs

Citrate-AuNPs and DMAP-AuNPs were prepared according to a procedure reported in the literature [29,30]. Their sizes and concentrations were determined according to the UV–Vis spectra, as measured using a spectrophotometer (UV-2400PC, Shimadzu, Kyoto, Japan) [31]. The sizes of the nanoparticles were estimated to be 30 nm for both citrate- and DMAP-AuNPs. The concentrations of citrate- and DMAP-AuNP solutions were determined to be 278 pM and 2.41 nM, respectively. Different sizes of DMAP-AuNPs were synthesized by varying the amount of sodium borohydride. The resultant DMAP-AuNP solutions were 5–10 nm for the concentration of 15.27 nM and 16–18 nm for the concentration of 122.6 nM, respectively.

## 2.4. Experimental set-up

The experimental set-up appears in figure 1. An Nd:YAG laser (532 nm, maximum power, 35 mW, Z40M18B-F-532-pz, Z-LASER, Germany) was focused loosely using a lens (focal distance, 50 mm, quartz) and was introduced into a cuvette placed on the stage of a microscope (Eclipse TE2000-S, Nikon, Japan). The size of the laser spot at the focal point was estimated to be 5 μm from the image of the fluorescence for a rhodamine B solution. The vesicle acceleration was monitored using a CCD camera (WAT-221S, Watece, NY, USA) interfaced with a PC via a TV capture board (29.97 fps, PCast, PC-MV5DX/U2, Buffalo, Aichi, Japan). A notch filter (Techapec rugate notch filter, 532 nm, optical density >4, stock number #46−565, Edmund Optics, NJ, USA) was placed in front of the CCD camera to exclude the scattered light from the laser beam. The vesicle was accelerated in the laser beam by optical force, and travelled several μm. After passing through the focal point, liquid resistance slowed the vesicle, and it eventually disappeared from the monitoring frame due to thermal convection. Therefore, the average velocity was calculated from the travelling distance and the time required for travelling across the monitoring frame. The average velocities were measured for more than four different vesicles under each condition.

# 3. Results and discussion

## 3.1. Enhanced acceleration of vesicles in the presence of the AuNPs

To clarify whether AuNPs could enhance the acceleration of vesicles, the velocity of the vesicles was measured in solutions with different concentrations of AuNPs. A video of a moving vesicle is given in the Dryad Digital Repository: https://doi.org/10.5061/dryad.tg60c4s (Movie-S1). Figure 2 shows the velocity for three types of vesicles with neutral, positive and negative charges in the presence of anionic citrate-AuNPs (*a*) and cationic DMAP-AuNPs (*b*). As seen in figure 2*a*, the cationic vesicles moved fastest in the presence of citrate-AuNPs because of an electrostatic interaction that was stronger

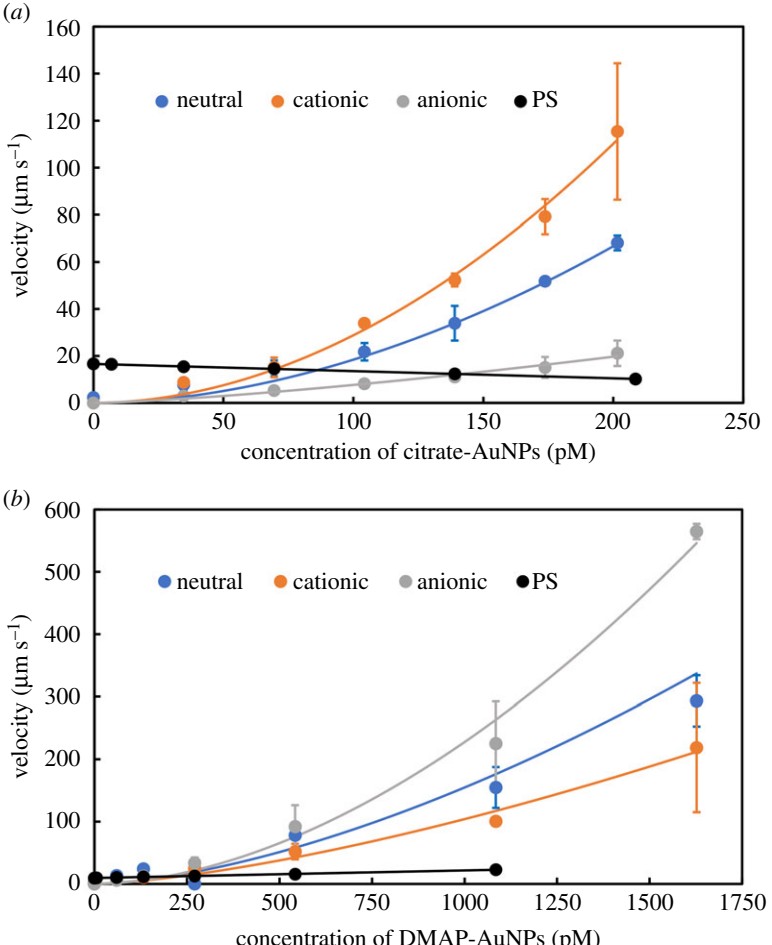

**Figure 2.** Enhancement of the velocity of the vesicle by increasing the concentration of the AuNPs. (*a*) Citrate-AuNPs, (*b*) DMAP-AuNPs. Laser power, 35 mW; medium, PBS. Vesicles, neutral, DPPC; cationic, DPPC:TAP = 10 : 1; anionic, DPPC:PA = 10 : 1; PS, polystyrene particle (1 μm). The sizes of the vesicles were adjusted to be 1 μm by passing through an extruder.

than that of the anionic and neutral vesicles. As expected, among the three types of vesicles, the cationic DMAP-AuNPs provided the most efficient acceleration of the anionic vesicles (figure 2*b*). Figure 2*a*,*b* clearly suggests that the velocity of the vesicles was efficiently increased when their charges were opposite that of the AuNPs, which was caused by the electrostatic interactions of the vesicles with AuNPs. Regardless of the expectations for electric repulsion, however, both the neutral vesicles and those with the same charge as the AuNPs increased in velocity in the presence of AuNPs. This phenomenon can be explained by considering that the charged phospholipids made up only 10% of the total phospholipid content in this experiment, because hydrophobic interactions still attracted AuNPs onto the vesicles. Therefore, increases in the velocities of the vesicles with the same charge as the AuNPs were due to the hydrophobic interaction between the neutral phospholipids of the charged vesicles and the charged AuNPs.

To confirm that an interaction with AuNPs leads to an increase in the velocity of vesicles, polystyrene particles were also accelerated by optical force in the presence of AuNPs. Interestingly, citrate-AuNPs slightly decreased the velocity of the polystyrene particles whereas DMAP-AuNPs caused a slight increase in the velocity (figure 2). The dependence that AuNP concentration exerted on velocity obviously differed from the effect that vesicles exerted. For example, the velocity of the polystyrene particle was increased by a factor of 1.5, at 1080 pM for DMAP-AuNPs and was decreased by a factor of 0.6 at 209 pM for citrate-AuNPs. Conversely, under the same conditions, the velocities of neutral vesicles were increased 150-fold and 30-fold in the presence of DMAP- and citrate-AuNPs, respectively. These results show that the polystyrene particles had a much weaker, or no, interaction with the AuNPs, which was independent of their charges.

A small decrease in the velocity of the polystyrene particle by the anionic AuNPs was due to the scattering of the laser light by the free AuNPs dispersed in the solution. The scattering of the AuNPs

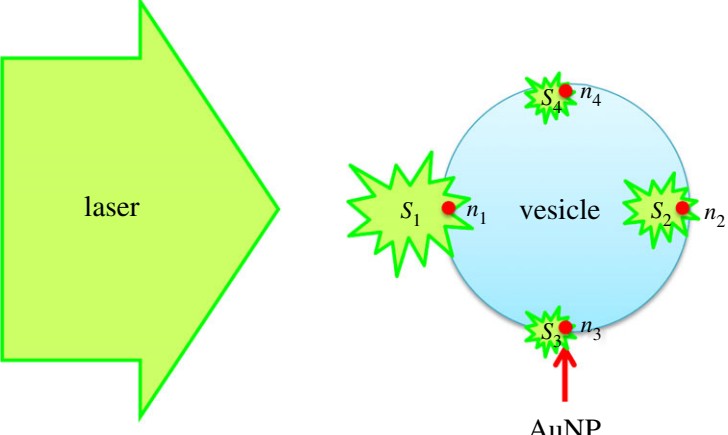

**Figure 3.** Schematic illustration of the model.

reduced the transmittance of the light, which resulted in a decrease in the light intensity irradiating the polystyrene particle, i.e. a decreased optical force. Conversely, as indicated by the slight increase in velocity, the polystyrene particles seemed to interact with cationic AuNPs, although the effect was much weaker than that in the vesicles.

The results showed that the AuNPs have less interaction with polystyrene particles. This suggests that direct collisions with AuNPs have no significant impact on the acceleration of particles and vesicles. Furthermore, the results of the vesicles with different charges indicated that both hydrophobic and electrostatic interactions play significant roles in the acceleration of the vesicles. Therefore, the surface charges of both the vesicles and the AuNPs are important factors in the enhancement of the acceleration of the vesicles.

## 3.2. Adsorption model of AuNPs on vesicles

The velocity of a vesicle is expected to correlate with the amount of AuNPs that are bound to it. Therefore, we proposed a model that could be used to estimate the binding constant and the apparent number of AuNPs adsorbed on a vesicle. A schematic illustration of the proposed model is shown in figure 3. In this illustration, we considered a case whereby four AuNPs could bind with a vesicle. Individual AuNPs are expected to bind at different sites on the vesicles, so the degrees of the scattering forces generated by the AuNPs depend on the location relative to the propagation direction of the laser beam. In figure 3, we assumed that AuNPs, $n_1$, $n_2$, $n_3$ and $n_4$ enhanced the light scattering at relative efficiencies of $S_1$, $S_2$, $S_3$ and $S_4$, respectively. Then, the total enhancement of the scattering force would be expected to be related to the $S$ values.

In the model, the binding constant, $K$, between the vesicle (Ve) and the AuNPs (NP), was defined as shown in equation (3.1).

$$K = \frac{[\mathrm{Ve(NP)}_n]}{[\mathrm{Ve}][\mathrm{NP}]^n}. \tag{3.1}$$

In equation (3.1), $[\mathrm{Ve(NP)}_n]$ is the concentration of vesicles bound with AuNPs, $[\mathrm{Ve}]$ is the concentration of free vesicles and $[\mathrm{NP}]$ is the concentration of the free AuNPs. It should be noted that $n$ is the apparent number because the AuNPs used in this study had polydispersed sizes which influence the efficiency of the enhancement of the scattering force. Mass balances of the vesicles and AuNPs are given in equations (3.2) and (3.3).

$$C_{\mathrm{Ve}} = [\mathrm{Ve}] + [\mathrm{Ve(NP)}_n] \tag{3.2}$$

and
$$C_{\mathrm{NP}} = [\mathrm{NP}] + n[\mathrm{Ve(NP)}_n]. \tag{3.3}$$

In equations (3.2) and (3.3), $C_{\mathrm{Ve}}$ is the total concentration of a vesicle and $C_{\mathrm{NP}}$ is the total concentration of the AuNPs. Applying equations (3.2) and (3.3) to equation (3.1), gives equation (3.4).

$$K\left(C_{\mathrm{Ve}} - \frac{C_{\mathrm{NP}} - [\mathrm{NP}]}{n}\right) = \frac{(C_{\mathrm{NP}} - [\mathrm{NP}])/n}{[\mathrm{NP}]^n}. \tag{3.4}$$

**Table 1.** Parameters obtained by fitting the proposed model. The molecularly occupied area of 55 Å$^2$ molecule$^{-1}$ for all phospholipids was employed in the calculation of $C_{Ve}$ [33].

| vesicle | $C_{Ve}$ (pM) | AuNPs | $K/a$ (M$^{-(n+1)}$ μm s$^{-1}$) | $n$ | $R^2$ |
|---|---|---|---|---|---|
| anionic | 12.4 | anionic | $1.53 \times 10^{25}$ | 1.35 | 0.983 |
| | | cationic | $2.98 \times 10^{30}$ | 1.95 | 0.995 |
| neutral | 11.8 | anionic | $3.12 \times 10^{30}$ | 1.86 | 0.981 |
| | | cationic | $1.19 \times 10^{27}$ | 1.57 | 0.973 |
| cationic | 13.7 | anionic | $2.42 \times 10^{31}$ | 1.93 | 0.993 |
| | | cationic | $6.91 \times 10^{25}$ | 1.46 | 0.991 |

In this study we assumed that the velocity of the vesicle was directly proportional to $C_{NP} - [NP]$ which indicated the concentration of the AuNPs bound to the vesicle, as shown in equation (3.5).

$$C_{NP} - [NP] = av. \tag{3.5}$$

In equation (3.5), $a$ is a constant to relate $C_{NP} - [NP]$ with $v$. Then, equation (3.4) can be rewritten as shown in equation (3.6).

$$av = \frac{KC_{Ve}\,n[NP]^n}{1 + K[NP]^n}. \tag{3.6}$$

In equation (3.6), the velocity reaches a plateau ($av = C_{Ve}n$) if $K[NP]^n$ is much larger than 1. When $K[NP]^n$ is much smaller than 1 and $C_{Ve}$ is much smaller than $C_{NP}$, $[NP]$ approximates $C_{NP}$, and equation (3.6) can be rewritten, as shown in equation (3.7).

$$v = \frac{K}{a}C_{Ve}nC_{NP}{}^n. \tag{3.7}$$

It is interesting that equation (3.7) is quite similar to the following Freundlich adsorption isotherm [32], which is shown here as equation (3.8).

$$\theta = kC^{1/n}. \tag{3.8}$$

In equation (3.8), $\theta$ is the molar (or weight) adsorbed amount per unit weight of adsorbent, $C$ is the molar (or weight) concentration of adsorbate in solution at equilibrium, and $k$ and $n$ are the experimental parameters that depend on the system of adsorbent and adsorbate. The Freundlich adsorption isotherm is an empirical relationship, and the meaning of $k$ and $n$ are unclear. Conversely, in the proposed model, the exponential term $n$ indicates the apparent number of the AuNPs adsorbed on the vesicle, and $K$ is the binding constant between the vesicles and the AuNPs. Therefore, our model would be more useful and meaningful than the Freundlich adsorption isotherm in this study because the binding constant and the apparent adsorption number can be estimated using the model.

According to equation (3.7), $K/a$ and $n$ were estimated using the results in figure 2. It should be noted that $n$ is the apparent number of AuNPs involved in the enhancement of scattering. Namely, the apparent number, $n$, is related to the enhancement of the scattering light when the maximum number of AuNPs is bound to the vesicles, but is not the total number of AuNPs on the vesicles. The regression curves are shown in figure 2, and the estimated values of $K/a$ and $n$ are given in table 1, where the molecularly occupied area of 55 Å$^2$ molecule$^{-1}$ was employed for all phospholipids in the calculation of $C_{Ve}$ [33]. The regression curves based on the model are in good agreement with the experimental data, as shown by the correlation coefficients. The binding constants indicated strong interactions between the pairs where the vesicle and the AuNPs had opposite charges. Surprisingly, the binding constant of the neutral vesicles with the anionic AuNPs was as large as that of the pairs with opposite charges. The same tendency was found in the apparent numbers of the AuNPs binding to the vesicles, i.e. the value was largest for the pairs with opposite charges and the pairs made up of neutral vesicles and anionic AuNPs. The decimal values of $n$ can be explained by accounting for the dispersion in the size of the AuNPs and different efficiencies of light scattering depending on the binding sites of individual AuNPs. The resonant wavelength of AuNPs depended on their size. Laser light was scattered

efficiently when the resonant wavelength of the AuNPs matched the emission wavelength of the laser. Therefore, the individual particles in the polydispersed AuNPs showed different scattering efficiencies, which means the decimal values are reasonable in our model. In addition, individual AuNPs at different binding sites showed different scattering efficiencies, which resulted in the decimal values of $n$.

As table 1 shows, the $n$ values were roughly 1.9 for the pair of the anionic vesicles and the cationic AuNPs, the pair of the cationic vesicles and the anionic AuNPs, and the pair of neutral vesicles and the anionic AuNPs. Conversely, when the vesicles and the AuNPs had the same charge, the $n$ values were roughly 1.5. To avoid misunderstanding of the model, we should emphasize that the $n$ values mean the apparent number that relates to the efficiency of the enhanced scattering force, rather than the actual number of AuNPs on the vesicle. The large values of the binding constant and the $n$ values for the pair with the opposite charges indicate that the hydrophobic and electrostatic attractions are dominant interactions between the vesicles and the AuNPs. The reason for the strong interaction between the neutral vesicles and the anionic AuNPs is unclear and rather strange because the neutral vesicles were slightly anionic in PBS [34,35]. Therefore, hydrogen bonding may play a role in the binding, although it is difficult to clarify the effect. For the pair with the same charge, a weak interaction is expected when the electrostatic repulsion is taken into account. At this point, it is important to reiterate that the charged vesicles attracted AuNPs of a like charge by hydrophobic interaction, because only 10% of the neutral phospholipids were replaced with charged phospholipids.

The enhanced acceleration can definitely be attributed to an increase in the optical force acting on the vesicles. Optical force acting on an object, $F$, is given by equation (3.9) [36].

$$F = Q_s \frac{n_1 P}{c}. \tag{3.9}$$

In equation (3.9), $Q_s$ is a coefficient representing the conversion efficiency of the light to the scattering force, $n_1$ is the refractive index of the surrounding medium, $P$ is the incident power and $c$ is the velocity of light. When a sphere travels in a solution, the velocity of the vesicles can be written as in equation (3.10) [37].

$$Q_s \frac{n_1 P}{c} = 6\pi\eta dv. \tag{3.10}$$

In equation (3.10), $\eta$ is the viscosity of the medium and $d$ is the diameter of the sphere. Therefore, we speculated that the enhanced velocity of the vesicle was due to the increased $Q_s$ value induced by the enhanced scattering of the AuNPs on the surface of the vesicles. We concluded that the AuNPs adsorbed on the vesicles enhanced the scattering of the laser light, which resulted in an increase in the optical force.

To clarify the dependence of the velocity on the size of the AuNPs, we synthesized different sizes of AuNPs with diameters of 5–10, 16–18 and 30 nm. Using the neutral vesicles, the velocity was measured in the presence of each of the DMAP-AuNPs at different concentrations. The velocity of the vesicle increased exponentially as the diameter of the AuNPs increased, as shown in figure 4. The curves represent the fitting results according to equation (3.7). In figure 4a, the 30 nm AuNPs enhanced the velocity of the vesicles more significantly than the smaller AuNPs. The $K/a$ obtained by the fitting were almost constant ($1.18 \times 10^{27}$ for 5–10 nm, $1.19 \times 10^{27}$ for 16–18 nm, and $1.19 \times 10^{27}$ for 30 nm) whereas the $n$ values decreased slightly with increase in the size of AuNPs (1.84 for 5–10 nm, 1.74 for 16–18 nm, 1.57 for 30 nm). The different $n$ values imply that the relative efficiency of light scattering depends on the size of the AuNPs because of their different extinction coefficients.

It was interesting when the horizontal axis of the concentration was replaced with the extinction of the solution, as shown in figure 4b. Figure 4b illustrates the linear relationship between the extinction and the velocity. The 30 nm AuNPs scattered laser light more efficiently than the smaller ones both in the solution and on the vesicle because of its large extinction coefficient. However, the free AuNPs in the solution weakened the intensity of the laser light via the enhancement of light scattering, which resulted in a decrease in the velocity of the vesicle. Therefore, this implies that the velocity of the vesicle is related to the light intensity scattered by the adsorbed AuNPs. These results also support the validity of the model where the AuNPs adsorbed on the vesicles enhanced the scattering of the laser light, which resulted in an increase in the optical force.

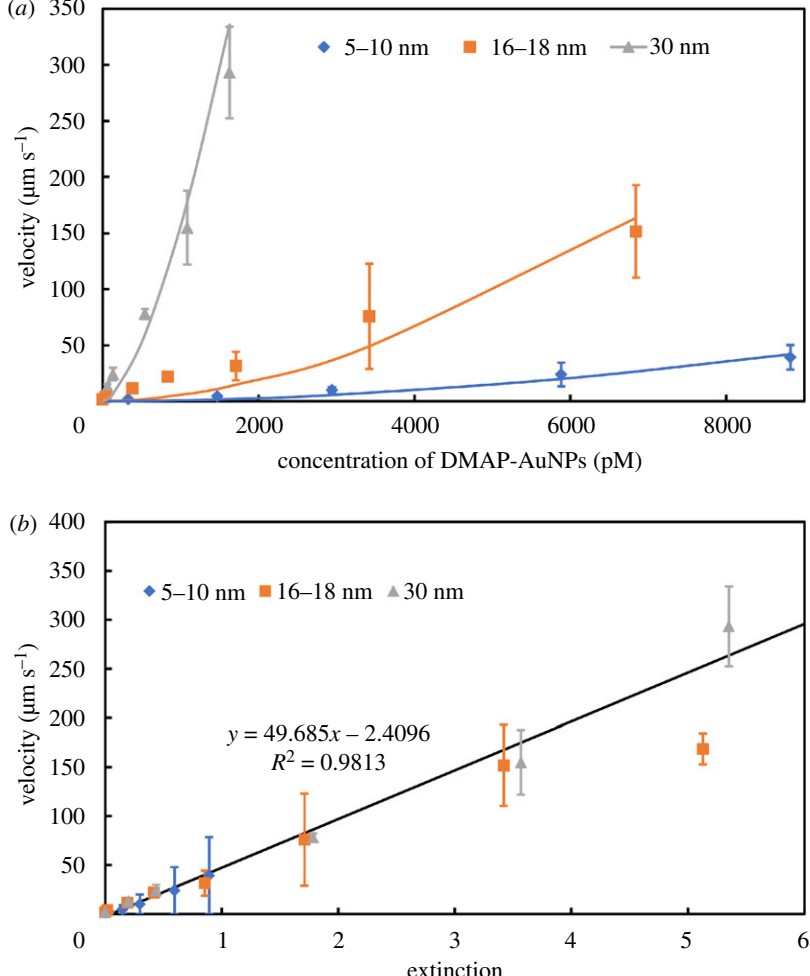

**Figure 4.** Enhancement in the velocity of the vesicles with different sizes of AuNPs. (*a*) The relationship between the concentration of the AuNPs and the velocity of the vesicle. (*b*) The relationship between the extinction of the AuNPs and the velocity of the vesicle.

## 4. Conclusion

In this study, the acceleration of vesicles via optical force was enhanced when AuNPs were adsorbed on their surface. Three types of vesicles (neutral, positive and negative) were used. Vesicles with positive and negative charges were efficiently accelerated in the presence of anionic citrate-AuNPs and cationic DMAP-AuNPs, respectively. The maximal numbers of adsorbed AuNPs were achieved when anionic vesicles were paired with cationic AuNPs, cationic vesicles were paired with anionic AuNPs and neutral vesicles were paired with anionic AuNPs. The apparent adsorption number of AuNPs was reduced when the vesicle charge was the same as that of the AuNPs, which was due to electric repulsion. These results imply that hydrophobic and electrostatic interactions play important roles in adsorption. The present study established that the enhanced collection efficiency induced by AuNPs was caused by thermal convection and acceleration of the vesicles due to the adsorption of AuNPs via electrostatic and hydrophobic interactions. The enhanced acceleration of the vesicles can be attributed to the strong optical force generated by the enhanced light scattering from the AuNPs on the surface of a vesicle. We also proposed a model that can be used to estimate the relative magnitude of the binding constant and the apparent number of the AuNPs adsorbed on the vesicles. According to these results, control of the surface charge on the AuNPs and increasing the concentration of AuNPs enhanced the collection efficiency of the vesicles and was applicable to the selective isolation of small vesicles, which included the extracellular examples.

Ethics. All experiments were conducted in a laboratory at the Department of Chemistry, Graduate School of Natural Science and Technology, Okayama University. All experimental results included in this paper were tested repeatedly and confirmed to be repeatable.

Data accessibility. All the experimental data are included in the manuscript. The data are available in the Dryad Digital Repository at: https://doi.org/10.5061/dryad.tg60c4s [38].

Authors' contributions. Y.T. and T.K. designed the experiments, carried out the experiments, wrote this manuscript, and created the figures. T.K. supervised the research. All authors discussed the results and commented on the manuscript.

Competing interests. We declare we have no competing interests.

Funding. This research was supported by JSPS KAKENHI grant nos. 17H05465 and 19H04675.

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
