## [Reviewer comments · Royal Society Open Science]

Review History

RSOS-190293.R0 (Original submission)

Review form: Reviewer 1

Is the manuscript scientifically sound in its present form?

Yes

Are the interpretations and conclusions justified by the results?

No

Is the language acceptable?

Yes

Is it clear how to access all supporting data?

Not Applicable

Do you have any ethical concerns with this paper?

No

Have you any concerns about statistical analyses in this paper?

Yes

Recommendation?

Major revision is needed (please make suggestions in comments)

Comments to the Author(s)

Comments to the Author

This paper describes that the optical force driven movement of vesicles was accelerated in the presence of gold nanoparticles (AuNPs). The author concluded that the acceleration was caused by AuNPs adsorbed on the vesicles. However, I do not wholly agree with the authors' model. I recommended that the paper should not be published until all questions listed below are answered satisfactorily.

- 1) The size of the vesicles used in this experiment should be described.
- 2) To demonstrate the acceleration of the movement of vesicles by light pressure, I strongly encourage the author to show the photographs or video that is the basis for calculating the velocity of vesicles.
- 3) The velocity (v) of an individual vesicle might be expected to be proportional to the number (n) of AuNPs adsorbed on its surface. However, the authors assumed that the velocity of the vesicle was proportional to $n[\text{Ve}(\text{NP})^{_n}]$. The physical meaning of equation (5) is incomprehensible for me.
- 4) On page 10, line 45; The meaning of “ a is a constant to relate W with v .” is unclear, because “ W ” is not defined.
- 5) The n values listed in Table 1 are less than 2. Therefore, the authors predicted that one or two AuNPs were bound to one vesicle. If the movement of vesicles by light pressure is governed by the AuNPs, the velocity of the vesicles is expected to be a maximum when two AuNPs adhere to the surface. However, in Fig. 2, the velocity of vesicles increases exponentially with the increase of the concentration of AuNPs. Therefore, the value of n obtained by curve fitting using eq. 7 seems to be physically meaningless. It is necessary to explain why the velocity of vesicles does not have the maximum value.
- 6) The n values indicated that one or two AuNPs were bound to one vesicle. Under the such conditions, the velocity of vesicles should show discrete values. Consequently, even at the same concentration of AuNPs, the velocity of individual vesicles is expected to be categorized into three different types (i.e., without AuNPs, one AuNP, two AuNPs). Since only four vesicle speeds were measured in this experiment, it is doubtful that statistically significant velocities have been estimated.
- 7) The data of Fig.3(a) should also be curve fitted by eq. 7, and the fitting parameters should be discussed in comparison with the values in Table 1.
- 8) Figure 3(b) represents a linear relationship between the extinction and the velocity. The authors claim that the velocity of the vesicle is directly proportional to the light intensity scattered by the adsorbed AuNPs. However, if an excess amount of free AuNPs exists in the solution, the extinction of the solution would be mainly ascribed to the free AuNPs. The explanation for Figure 3(b) seems to be inconsistent with the conclusions.

Review form: Reviewer 2

Is the manuscript scientifically sound in its present form?

Yes

Are the interpretations and conclusions justified by the results?

Yes

Is the language acceptable?

Yes

Is it clear how to access all supporting data?

Not Applicable

Do you have any ethical concerns with this paper?

No

Have you any concerns about statistical analyses in this paper?

No

Recommendation?

Accept with minor revision (please list in comments)

Comments to the Author(s)

This manuscript is reporting the enhancement of optical force acting on vesicles via the binding of AuNPs. The authors discussed in detail by controlling the surface charges in the vesicles and the AuNPs, and by changing the sizes of the AuNPs. Also, the authors proposed a model that could be used to estimate the binding constant and the apparent number of AuNPs adsorbed on a vesicle. The study is systematic and interesting. In my opinion, this manuscript is acceptable for publication, but a minor revision should be implemented to the manuscript before its publication in RSOS.

1. P7 L9-14: The frame rate of the present system should be provided. In addition, a captured image should be provided if the authors have recorded.

2. P10 L 45: "a is a constant to relate W with v." What does this "W" mean?

3. "The molecularly occupied area of 55 A² molecule⁻¹ for all phospholipids was employed in the calculation of C_{ve}.33 (Table 1)" and "The sizes of the vesicles were adjusted to be 1 μm by passing through an extruder. (Fig. 1)" These are very important information. Therefore, these sentences should be (also) described in the main body of the text.

Decision letter (RSOS-190293.R0)

19-Mar-2019

Dear Dr Kaneta:

Title: Enhancement of Optical Force Acting on Vesicles via the Binding of Gold Nanoparticles
Manuscript ID: RSOS-190293

The editor assigned to your manuscript has now received comments from reviewers. We would like you to revise your paper in accordance with the referee and Subject Editor suggestions which can be found below (not including confidential reports to the Editor). Please note this decision does not guarantee eventual acceptance.

Please submit your revised paper before 11-Apr-2019. Please note that the revision deadline will expire at 00.00am on this date. If we do not hear from you within this time then it will be assumed that the paper has been withdrawn. In exceptional circumstances, extensions may be possible if agreed with the Editorial Office in advance. We do not allow multiple rounds of revision so we urge you to make every effort to fully address all of the comments at this stage. If deemed necessary by the Editors, your manuscript will be sent back to one or more of the original reviewers for assessment. If the original reviewers are not available we may invite new reviewers.

- Acknowledgements

RSC Associate Editor:

Comments to the Author:
(There are no comments.)

RSC Subject Editor:
Comments to the Author:
(There are no comments.)

Reviewers' Comments to Author:
Reviewer: 1

Comments to the Author(s)
Comments to the Author

This paper describes that the optical force driven movement of vesicles was accelerated in the presence of gold nanoparticles (AuNPs). The author concluded that the acceleration was caused by AuNPs adsorbed on the vesicles. However, I do not wholly agree with the authors' model. I recommended that the paper should not be published until all questions listed below are answered satisfactorily.

- 1) The size of the vesicles used in this experiment should be described.
- 2) To demonstrate the acceleration of the movement of vesicles by light pressure, I strongly encourage the author to show the photographs or video that is the basis for calculating the velocity of vesicles.
- 3) The velocity (v) of an individual vesicle might be expected to be proportional to the number (n) of AuNPs adsorbed on its surface. However, the authors assumed that the velocity of the vesicle was proportional to $n^{1/2}$. The physical meaning of equation (5) is incomprehensible for me.
- 4) On page 10, line 45; The meaning of " a is a constant to relate W with v ." is unclear, because " W " is not defined.
- 5) The n values listed in Table 1 are less than 2. Therefore, the authors predicted that one or two AuNPs were bound to one vesicle. If the movement of vesicles by light pressure is governed by the AuNPs, the velocity of the vesicles is expected to be a maximum when two AuNPs adhere to the surface. However, in Fig. 2, the velocity of vesicles increases exponentially with the increase of the concentration of AuNPs. Therefore, the value of n obtained by curve fitting using eq. 7 seems to be physically meaningless. It is necessary to explain why the velocity of vesicles does not have the maximum value.
- 6) The n values indicated that one or two AuNPs were bound to one vesicle. Under the such conditions, the velocity of vesicles should show discrete values. Consequently, even at the same concentration of AuNPs, the velocity of individual vesicles is expected to be categorized into three different types (i.e., without AuNPs, one AuNP, two AuNPs). Since only four vesicle speeds were measured in this experiment, it is doubtful that statistically significant velocities have been estimated.
- 7) The data of Fig.3(a) should also be curve fitted by eq. 7, and the fitting parameters should be discussed in comparison with the values in Table 1.

8) Figure 3(b) represents a linear relationship between the extinction and the velocity. The authors claim that the velocity of the vesicle is directly proportional to the light intensity scattered by the adsorbed AuNPs. However, if an excess amount of free AuNPs exists in the solution, the extinction of the solution would be mainly ascribed to the free AuNPs. The explanation for Figure 3(b) seems to be inconsistent with the conclusions.

Reviewer: 2

Comments to the Author(s)

This manuscript is reporting the enhancement of optical force acting on vesicles via the binding of AuNPs. The authors discussed in detail by controlling the surface charges in the vesicles and the AuNPs, and by changing the sizes of the AuNPs. Also, the authors proposed a model that could be used to estimate the binding constant and the apparent number of AuNPs adsorbed on a vesicle. The study is systematic and interesting. In my opinion, this manuscript is acceptable for publication, but a minor revision should be implemented to the manuscript before its publication in RSOS.

1. P7 L9-14: The frame rate of the present system should be provided. In addition, a captured image should be provided if the authors have recorded.

2. P10 L 45: "a is a constant to relate W with v." What does this "W" mean?

3. "The molecularly occupied area of 55 A² molecule⁻¹ for all phospholipids was employed in the calculation of C_v.33 (Table 1)" and "The sizes of the vesicles were adjusted to be 1 μm by passing through an extruder. (Fig. 1)" These are very important information. Therefore, these sentences should be (also) described in the main body of the text.

Author's Response to Decision Letter for (RSOS-190293.R0)

See Appendix A.

Decision letter (RSOS-190293.R1)

15-Apr-2019

Dear Dr Kaneta:

Title: Enhancement of Optical Force Acting on Vesicles via the Binding of Gold Nanoparticles
Manuscript ID: RSOS-190293.R1

It is a pleasure to accept your manuscript in its current form for publication in Royal Society Open Science. The chemistry content of Royal Society Open Science is published in collaboration with the Royal Society of Chemistry.

Yours sincerely,

Dr Laura Smith
Publishing Editor, Journals

Appendix A

To the Reviewer: 1

Thank you very much for your helpful comments and suggestions. We carefully considered the comments raised by the reviewer. Our replies to the comments are as follows.

This paper describes that the optical force driven movement of vesicles was accelerated in the presence of gold nanoparticles (AuNPs). The author concluded that the acceleration was caused by AuNPs adsorbed on the vesicles. However, I do not wholly agree with the authors' model. I recommended that the paper should not be published until all questions listed below are answered satisfactorily.

1) The size of the vesicles used in this experiment should be described.

We also added a description in the text according to the suggestion by the reviewer. (P.6, L.7-8)

2) To demonstrate the acceleration of the movement of vesicles by light pressure, I strongly encourage the author to show the photographs or video that is the basis for calculating the velocity of vesicles.

We added a movie in the Supporting information as suggested by the reviewer. (P.7, L.18-19, at the Dryad Digital Repository, Movie S1)

3) The velocity (v) of an individual vesicle might be expected to be proportional to the number (n) of AuNPs adsorbed on its surface. However, the authors assumed that the velocity of the vesicle was proportional to $n[\text{Ve}(\text{NP})^n]$. The physical meaning of equation (5) is incomprehensible for me.

According to the mass balance of AuNPs as shown in equation (3), $n[\text{Ve}(\text{NP})^n]$ represents the concentration of AuNPs adsorbed on the vesicles. The number of AuNPs adsorbed on the vesicles also depends on the concentration of vesicles when considering binding equilibrium employed in our model. So, we assumed that the velocity of the vesicle is proportional to $n[\text{Ve}(\text{NP})^n] = \text{CNP}-[\text{NP}]$, which is the concentration of AuNPs bound to the vesicles. However, the expression made understanding difficult as found in the reviewer's comment, so we decided to delete the part of $n[\text{Ve}(\text{NP})^n]$ from the equation (5). (P.10, L.21-P.11, L.2)

4) On page 10, line 45; The meaning of “ a is a constant to relate W with v .” is unclear, because “ W ” is not defined.

We apologize for our mistake. The “ W ” must be replaced by $\text{CNP}-[\text{NP}]$. We corrected the mistake in the text. (P.11, L.3)

5) The n values listed in Table 1 are less than 2. Therefore, the authors predicted that one or two AuNPs were bound to one vesicle. If the movement of vesicles by light pressure is governed by the AuNPs, the velocity of the vesicles is expected to be a maximum when two AuNPs adhere to the surface. However, in Fig. 2, the velocity of vesicles increases exponentially with the increase of the concentration of AuNPs. Therefore, the value of n obtained by curve fitting using eq. 7 seems to be physically meaningless. It is necessary to explain why the velocity of vesicles does not have the maximum value.

We apologize for the insufficient explanation of our model. It should be noted that the n values shown in Table 1 are the “apparent” number of AuNPs bound to the vesicles. In the model, we do not assume that the velocity of an individual vesicle is proportional to the number (n) of AuNPs adsorbed on its surface because the light scattering induced by AuNPs depends on the adsorbing site of the AuNPs on the vesicle. Namely, the degree of the increase in the velocity of the vesicle depends on both the number and the adsorbing site of the AuNPs on the vesicle. We added a schematic illustration of our model in figure 3 along with a discussion to help readers understand the model. (P.9, L.21- P.10, L.6)

Conversely, the reason the velocity of vesicles does not have the maximum value is attributed to a low concentration of AuNPs that is insufficient to reach the maximum velocity. We added this explanation to the text. (P.11, L.6)

6) The n values indicated that one or two AuNPs were bound to one vesicle. Under the such conditions, the velocity of vesicles should show discrete values. Consequently, even at the same concentration of AuNPs, the velocity of individual vesicles is expected to be categorized into three different types (i.e., without AuNPs, one AuNP, two AuNPs). Since only four vesicle speeds were measured in this experiment, it is doubtful that statistically significant velocities have been estimated.

We agree with the reviewer’s suggestion in part. As suggested by the reviewer, it is incorrect that the maximum number of the AuNPs bound to the vesicle is 2. So, we deleted the description of “one or two AuNPs were bound to one vesicle.” to avoid misunderstanding by the readers. (P.2, L.18-20 and P.12, L.16-17)

Conversely, according to our model based on the binding equilibrium, the number of AuNPs bound to the vesicles depends on the concentration of the AuNPs added to the solution. As the reviewer pointed out, a few different vesicles with different numbers of AuNPs may exist in a solution. However, a major species would be stochastically observed at a certain concentration of the AuNPs.

Also, the reviewer was worried about whether the statistically significant velocities were estimated or not. We presented the standard deviations that suggest the errors of the data, so the experimental errors were considered in the results.

7) The data of Fig.3(a) should also be curve fitted by eq. 7, and the fitting parameters should be discussed in comparison with the values in Table 1.

We fitted the results in Fig.3 (a) to eq. 7 and discussed the values as suggested by the reviewer. (P.14, L.19-P.15, L.5)

8) Figure 3(b) represents a linear relationship between the extinction and the velocity. The authors claim that the velocity of the vesicle is directly proportional to the light intensity scattered by the adsorbed AuNPs. However, if an excess amount of free AuNPs exists in the solution, the extinction of the solution would be mainly ascribed to the free AuNPs. The explanation for Figure 3(b) seems to be inconsistent with the conclusions.

We apologize that our insufficient explanation caused a misunderstanding by the reviewer. Against the results in figure 3, free AuNPs existing in the solution should reduce the velocity of the vesicle because they will decrease the laser intensity that hits the vesicle. So, we cannot explain the acceleration of the vesicle if free AuNPs are not present in the laser light. The increased velocity of the vesicle is related to the enhanced light scattering, that is, the extinction, by the AuNPs that bind with the vesicle. To avoid the misunderstanding, we added a further explanation in the text. (P.15, L8-12)

To Reviewer: 2

Thank you very much for your helpful comments and suggestions. We carefully considered the comments raised by the reviewer. Our replies to the comments are as follows.

Comments to the Author(s)

This manuscript is reporting the enhancement of optical force acting on vesicles via the binding of AuNPs. The authors discussed in detail by controlling the surface charges in the vesicles and the AuNPs, and by changing the sizes of the AuNPs. Also, the authors proposed a model that could be used to estimate the binding constant and the apparent number of AuNPs adsorbed on a vesicle. The study is systematic and interesting. In my opinion, this manuscript is acceptable for publication, but a minor revision should be implemented to the manuscript before its publication in RSOS.

1. P7 L9-14: The frame rate of the present system should be provided. In addition, a captured image should be provided if the authors have recorded.

The frame rate of the present system should be provided. (P.7, L.4) Also, we added a movie in the Supporting information as suggested by the reviewer. (P.7, L.18-19, at the Dryad Digital Repository, Movie S1)

2. P10 L 45: “a is a constant to relate W with v.” What does this “W” mean?

We apologize for our mistake. The “W” must be replaced by CNP-[NP]. We corrected the mistake in the text. (P.11, L.3)

3. “The molecularly occupied area of 55 A² molecule⁻¹ for all phospholipids was employed in the calculation of C_v (Table 1)” and “The sizes of the vesicles were adjusted to be 1 μm by passing through an extruder. (Fig. 1)” These are very important

information. Therefore, these sentences should be (also) described in the main body of the text.

The description suggested by the reviewer was added to the text. (P.6, L.7-8 and P.12, L.7-9)